# Environmental circadian disruption re-writes liver circadian proteomes

Hao A. Duong [1,2] ✉, Kenkichi Baba[1,2], Jason P. DeBruyne [1,2], Alec J. Davidson [2], Christopher Ehlen [2], Michael Powell[3] & Gianluca Tosini [1,2]

Circadian gene expression is fundamental to the establishment and functions of the circadian clock, a cell-autonomous and evolutionary-conserved timing system. Yet, how it is affected by environmental-circadian disruption (ECD) such as shiftwork and jetlag are ill-defined. Here, we provided a comprehensive and comparative description of male liver circadian gene expression, encompassing transcriptomes, whole-cell proteomes and nuclear proteomes, under normal and after ECD conditions. Under both conditions, post-translation, rather than transcription, is the dominant contributor to circadian functional outputs. After ECD, post-transcriptional and post-translational processes are the major contributors to whole-cell or nuclear circadian proteome, respectively. Furthermore, ECD re-writes the rhythmicity of 64% transcriptome, 98% whole-cell proteome and 95% nuclear proteome. The re-writing, which is associated with changes of circadian regulatory cis-elements, RNA-processing and protein localization, diminishes circadian regulation of fat and carbohydrate metabolism and persists after one week of ECD-recovery.

The circadian clock is a cell-autonomous and evolutionary conserved timing system. It has evolved as a mechanism for organisms to synchronize daily cycles of internal biological rhythms with external environmental conditions. At the molecular level, the mammalian circadian clock system is established upon a transcription-translation feedback loop. In the activation phase of the loop, CLOCK/BMAL1 heterodimer activates the rhythmic expression of *Periods*, *Cryptochromes* and *Casein kinase*(s). The protein products of these genes form nuclear protein complex(es) that represses the transcriptional activation activity of CLOCK/BMAL1, closing the loop. This feedback loop drives daily rhythmic expression of thousands of clock-controlled genes, underlying the daily environmental-circadian synchronization[1–4].

Under environmental circadian disrupted conditions such as jetlag, social jetlag, and shiftwork, the natural harmonic alignment is disrupted due to perturbations of environmental conditions. Such perturbations resulted in an acceleration of neurological, cardiometabolic, and immune disorders and cancers[5–12]. This is a growing global

health challenge, affecting hundreds of millions of people worldwide. In the United States and European countries, about 20% of the workforce is subjected to such conditions via shiftwork, and the percentage is expected to be higher in developing countries. Yet how the circadian gene expression system is affected by these conditions remains unclear. To address this question as well as illuminate its molecular underpinnings and functional implications, we systematically interrogated changes in the circadian gene expression process—encompassing transcription (transcript abundance), translation (whole-cell protein abundance) and post-translation processing (nuclear protein abundance)—at the -omics level in livers of ECD (environmental circadian disruption) mice in comparison to those under standard circadian (STD) condition (Supplementary Fig. 1). ECD mouse is an established mouse model that broadly recapitulates chronic jetlag and shiftwork conditions[13].

Here we showed that ECD re-writes the rhythmicity, without changing the cumulative daily amount, of over 95% rhythmic proteins at the whole-cell or nuclear compartment level. The rhythmicity of 64%

[1]Department of Pharmacology and Toxicology, Morehouse School of Medicine, Atlanta, GA 30310, USA. [2]Department of Neurobiology, Morehouse School of Medicine, Atlanta, GA 30310, USA. [3]Department of Microbiology, Biochemistry and Immunology, Morehouse School of Medicine, Atlanta, GA 30310, USA. ✉e-mail: hduong@msm.edu

rhythmic transcripts as well as many circadian molecular functions were also re-written in response to ECD. Re-writing of transcript rhythmicity is associated with a change in the enrichment of circadian regulatory cis elements from BMAL1 to DEC1 binding sites, while re-writing of nuclear protein rhythmicity was associated with a change in rhythmic protein abundance of RAN and CAS, two key components of the nuclear protein import/export process. Additionally, we found post-translational processing, rather than transcription, is the dominant contributor to circadian protein rhythms at the compartmental level, where most functional outputs are performed, under both STD and ECD.

## Results

### Topography of circadian gene expression

The circadian gene expression process has been intensively investigated, resulting in a thorough understanding of circadian transcription regulation from chromatin opening, cis−trans regulator interaction to transcript abundance of core clock as well as thousands of clock-controlled genes[14–16]. However, our understanding of other subsequential stages beyond transcription such as post-transcription, translation and post-translation is sparse[17–20]. For example, while there have been multiple studies on circadian transcriptome[21,22] and circadian whole-cell proteome[17], no study was reported for nuclear circadian proteome, individually or together with a circadian transcriptome and a circadian whole-cell proteome. This piece of information is essential for partitioning the contribution of post-translation in the circadian gene expression process. To fill these gaps, we quantified the abundance of nuclear proteins, whole-cell proteins and transcripts in the same set of mouse livers at 8 time points (in triplicates) throughout a circadian cycle using quantitative mass spectrometry or RNA-seq analysis. These livers were collected under circadian (free running) condition starting on the 2nd day in constant darkness from male mice that were previously entrained to STD light cycle (Supplementary Fig. 1).

Within this set of mouse livers, we successfully quantified 21,061 transcript time series, and 7314 protein time series across whole-cell and nuclear compartments. To inquire the reliability of the dataset, we examined the patterns of transcript abundance of core clock genes in the circadian transcriptome. We found transcripts of all core clock genes including *Clock*, *Arntl*, *Per1/2/3*, *Cry1/2*, *Csnk1d/e* and *Nr1d1/2* with their patterns being consistent with previous studies[15,23]. Some of these patterns were validated by RT-qPCR (Fig. 1a-b, S2a; Supplementary Data 1 Sheets 1-3). The processing and results of mass spectrometry analyses were also validated. Dozens of known mouse liver nuclear proteins were found with more than 20 folds higher in the nuclear proteomes than in the whole-cell proteomes, indicating a proper enrichment of the nuclear extracts. Pearson analysis showed a strong correlation between proteomes from the same compartment and a negative correlation between proteomes from different compartments, consistent with proper handling of the samples. Furthermore, western blot analysis of whole-cell extracts showed circadian patterns of core clock proteins such as CLOCK, BMAL1 (ARNTL), PER2, and REV-ERBA (NR1D1) were similar to those in the whole-cell proteomes. These patterns are also consistent with published studies[24,25]. Lastly, gene ontology enrichment analysis showed the whole-cell circadian proteome was enriched with many known circadian functions such as endosome transport, autophagy and response to hypoxia as expected (Supplementary Fig. 2b−f). These results affirmed the reliability of the dataset.

Of 21,061 transcript time series, 5502 (26%) exhibited a rhythmic pattern of abundance with the peaks spreading throughout the cycle (Fig. 1c). Among the whole-cell protein time series, 285 (3.9%) were rhythmic. These proteins exhibited a bimodal distribution of peak abundance concentrating around CT7 and CT17. In the nuclear compartment, 678 (9.3%) were rhythmic with a distinct phase distribution

from that of the whole-cell circadian proteome, assuming an unimodal pattern centering around CT20 (Fig. 1c). Collectively, there were 6036 genes that exhibited a rhythmic pattern of abundance at the mRNA, whole-cell protein or nuclear protein level. Remarkably, only 22 genes showed rhythmicity at all three levels (Fig. 1b; Supplementary Data 1, sheet 4). This is a comprehensive and coherent description of the daily abundance of whole-cell transcripts, whole-cell proteins and nuclear proteins in the same set of tissues under the circadian condition. Such a dataset has not previously been described before.

### Differential rhythmicity of transcripts and proteins

The rhythmicity of a transcript abundance has been routinely used to infer circadian biological function, but it is a protein that mostly performs the gene function. We thus asked how faithfully transcript rhythmicity is recapitulated in protein rhythmicity. To minimize the effect of the difference in detection levels between RNA-seq and Mass Spectrometry on the analysis, we selected genes of which transcript(s) and protein(s) were quantified in the transcriptome and proteomes and then examined their relationship in time and space. We found 6937 of such genes, constituting 33% all quantified transcripts and 95% all quantified proteins (Fig. 1d, e; Supplementary Data 1, sheet 5). Within this sub-population, 3295 genes were rhythmic at the transcript or protein levels. More importantly, less than 5% rhythmic transcripts had rhythmic whole-cell proteins and 48% rhythmic whole-cell proteins had rhythmic transcripts (Fig. 1f, g). The percentage of rhythmic whole-cell proteins with rhythmic transcripts was close to those observed under diurnal condition[26,27]. These percentages indicated that most rhythmic transcripts do not associate with rhythmic proteins and the majority of rhythmic proteins do not directly originate from rhythmic transcripts although both forms of the gene products were detected. They also suggested that post-transcription contributes as much as transcription to circadian whole-cell protein rhythms.

Accordingly, comparative gene ontology analysis between rhythmic transcripts and whole-cell proteins, either top 285 or top 3.9% rhythmic time series, showed a stark difference in enriched terms between the populations. Highest enriched clusters such as nucleoside monophosphate metabolic process and early to late endosome transport were enriched in one rhythmic population but not the other (Fig. 1h). Furthermore, numerous genes that play important roles in key biological processes such as *Arntl/Bmal1* (circadian clock establishment), *Pgm2l1*, *Gtf2h2*, *Txndc9* (basal transcription) or (*Gc*, *Cyp4a14*, *Gpat4* (metabolism) also exhibited a difference in circadian rhythmicity between transcript and protein levels (Fig. 1i).

These results indicated circadian rhythmicity of transcripts does not closely reflect that of their functional form, proteins. This warrants extra caution in inferring a circadian function from rhythmic transcripts at individual or -omics levels.

### Post-translation underlies circadian nuclear proteome

Whole-cell protein abundance measures the collective amount of a protein in a cell but does not carry information on its distribution among dozens of subcellular organelles where cell functions are largely performed. To see how sub-cellular localizations, which are active and tightly regulated processes[28], contribute to circadian protein rhythmicity, thereby circadian functional outputs, we examined the overlap between whole-cell and nuclear circadian proteomes. These proteomes were from the same set of tissues and processed in parallel from tissue collection to mass-spec analysis. 94% of nuclear rhythmic proteins were not rhythmic at the whole-cell level (Fig. 1j). The rhythmicity of these proteins, which we refer to as GOR$_{PTA\_STD}$, must be acquired post-translationally as they were not rhythmic at the whole-cell level, making contribution from transcriptional, post-transcriptional and translational controls unlikely. Furthermore, only 14% of whole-cell rhythmic proteins could be allocated to the rhythmicity in the nuclear compartment (Fig. 1k). The remainder, 86%, had

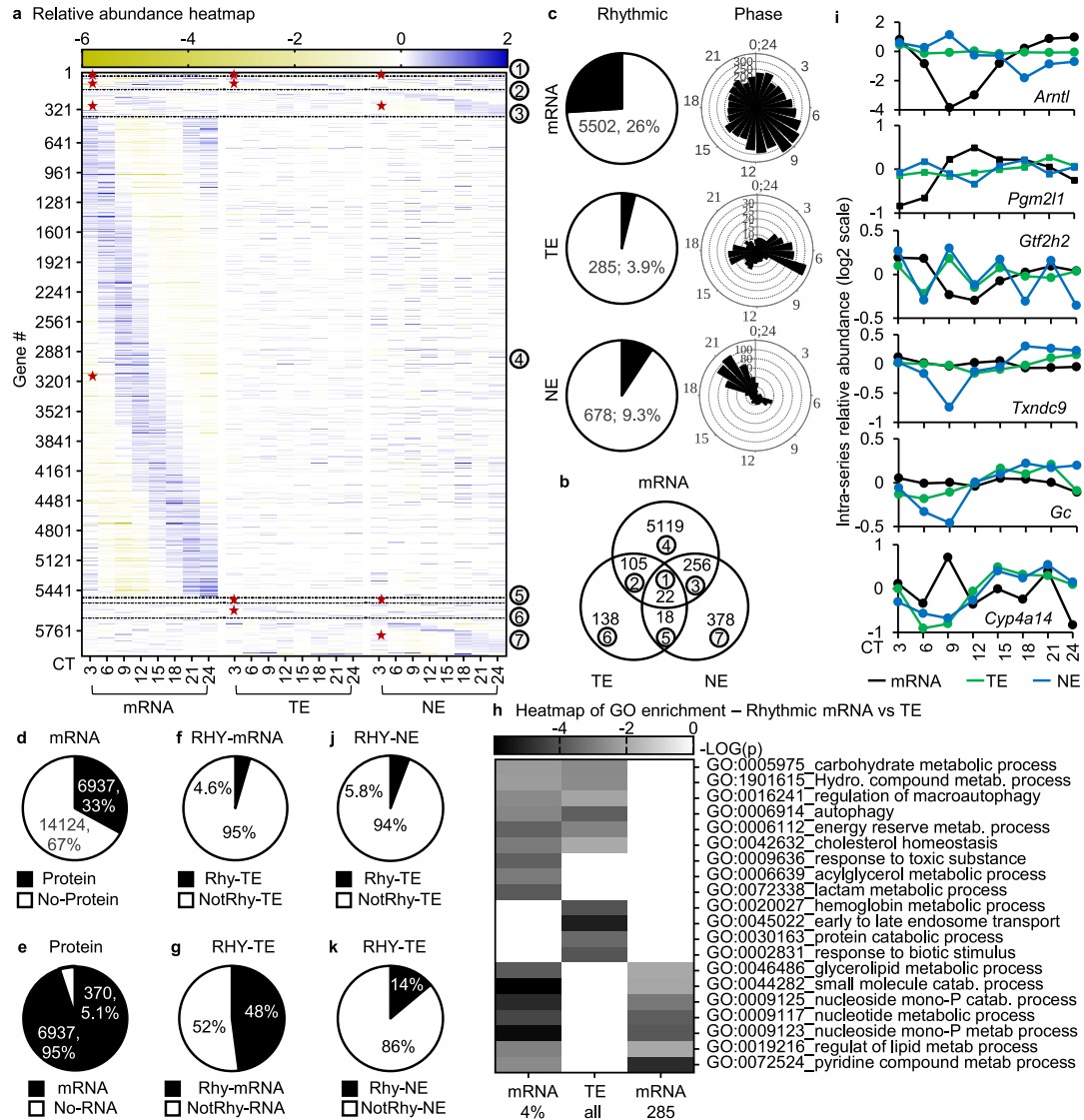

**Fig. 1 | A topography of circadian gene expression in mouse liver. a, b** Relative-abundance heatmap (normalized to series average; log$_2$ scale) of rhythmic transcripts (mRNA), rhythmic whole-cell proteins (TE) and rhythmic nuclear proteins (NE) (**a**), and the Venn diagram of their overlaps (**b**). **c** Percentages of rhythmic time series and their phase distribution of transcripts, whole cell proteins and nuclear proteins. **d–g** Percentage of transcripts with or without proteins (**d**), proteins with or without transcripts (**e**), rhythmic transcripts with or without rhythmic whole-cell proteins, (**f**) rhythmic whole-cell proteins with or without rhythmic transcripts (**g**) in the 6937 sub-population. **h** Comparative functional enrichment analysis of highest 285 or 4% rhythmic transcripts vs. whole-cell circadian proteome (285 proteins). **i** Examples of differential transcript, whole-cell protein and nuclear circadian patterns. **j, k** Percentages of rhythmic whole-cell proteins with or without rhythmic nuclear proteins (**j**), or rhythmic nuclear proteins with or without rhythmic whole-cell proteins (**k**) in the 6937 sub-population. TE−total extract; NE−nuclear extract; RHY−rhythmic; Red star−rhythmic portion; Number in circle−corresponding groups between (**a**) and (**b**).

to be attributed to rhythmicity in other compartments, re-constituting their rhythmicity at the whole-cell level. These results indicated nuclear circadian proteome arises post-translationally. They also suggested the existence of compartmental specific circadian proteomes, a less understood area in the field, not only in the nucleus but also in other organelles.

## ECD re-writes the circadian transcriptome
To examine the effect of ECD on the circadian gene expression process at the transcriptional level, we performed circadian RNA-seq analysis of mouse livers after ECD and then followed by a comparative analysis of its rhythmic transcript population with that under STD. After ECD we quantified 21,061 transcript time series and found 3445 (16%) transcripts exhibiting a circadian rhythmic pattern of abundance (Supplementary Data 2, Sheet 1). Transcripts of all core clock genes were found, and their patterns were validated by RT-qPCR

(Supplementary Fig. 3a). Among these rhythmic transcripts, 1061 (30.8%) gained their rhythmicity (GOR$_R$) in response to ECD as they were not rhythmic under STD. For the 2384 remaining transcripts, which retained their rhythmicity from STD to ECD (ROR$_R$), 504 displayed a phase shift that is greater than the 3-hr temporal resolution of sampling. Interestingly, all but 1 of these 504 ROR transcripts exhibited a phase advance coinciding with the shifted direction of the light cycle for ECD. Among 5502 STD-rhythmic transcripts, 3118 (56.7%) also lose their rhythmicity (LOR$_R$) in response to ECD. Collectively, there were 6,563 rhythmic transcripts, of which 72% changed rhythmicity comprising of 64% re-writing and 8% phase-advance in response to ECD. These rhythmic populations exhibited a distributed pattern of peak abundance throughout the cycle (Fig. 2a–d; Supplementary Data 2, Sheets 2-4). Among core clock transcripts, the rhythmicity of *Arntl*, *Per1/2*, *Cry1*, and *Nr1d1* showed mild perturbation in phase or amplitude, but none possessed a marked loss or gain of rhythmicity (Fig. 2e).

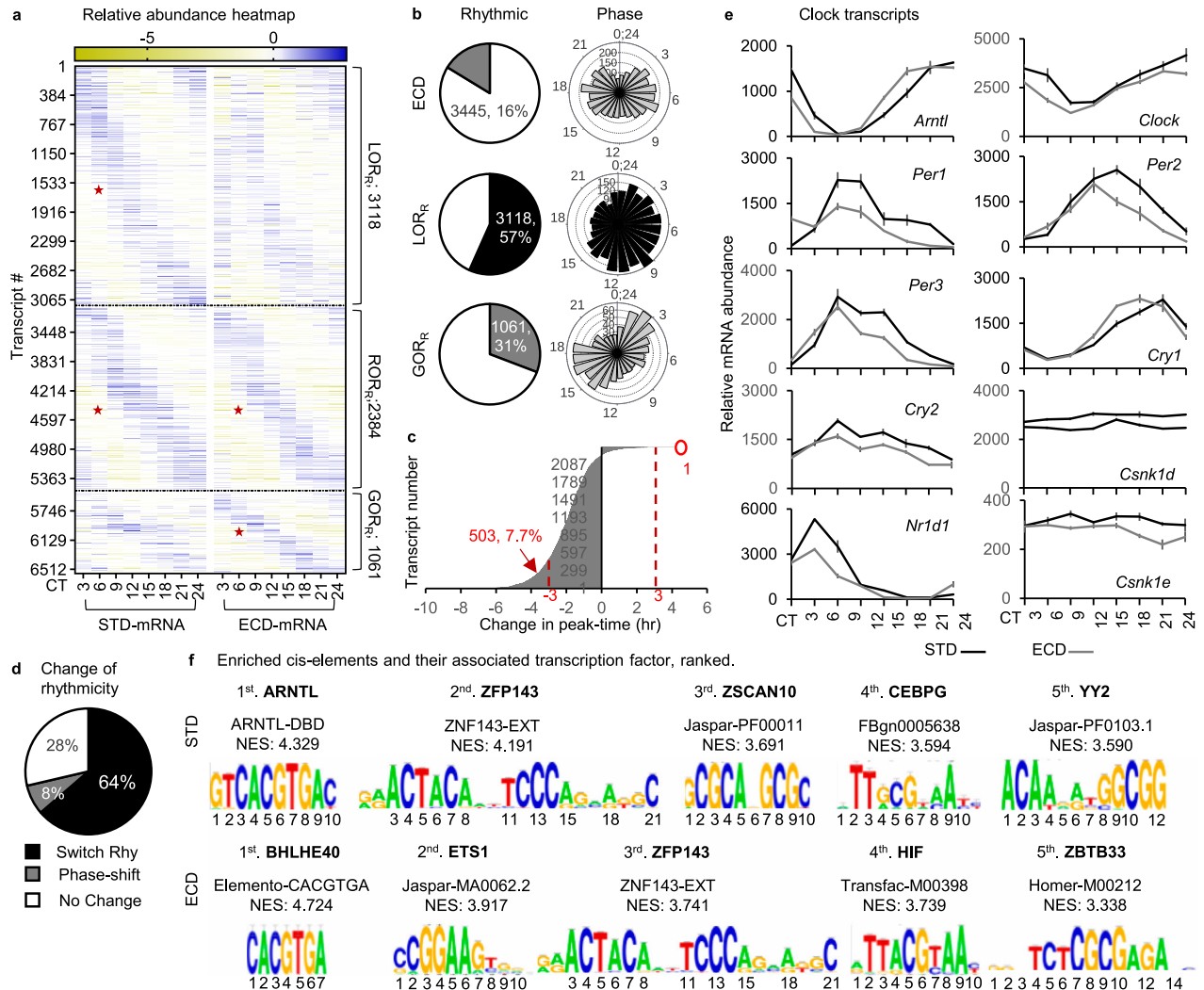

**Fig. 2 | ECD re-writes rhythmicity of 4179 transcripts. a** Relative abundance heatmap (normalized to series average; log₂ scale) of rhythmic transcripts in 21061 transcripts that were quantified under both standard (STD) and ECD conditions, grouping as rhythmic under STD but not under ECD (LOR$_R$), not rhythmic under STD but rhythmic under ECD (GOR$_R$) or rhythmic under both STD and ECD (ROR$_R$). **b** Percentages of rhythmic transcript time series and the corresponding phase distribution of transcripts that are either rhythmic under ECD, LOR$_R$ or GOR$_R$. **c** The distribution of phase shifts in the ROR$_R$ population between ECD and STD.

**d** Percentages of transcripts that switch their rhythmicity in response to ECD. **e** Comparative abundance of core clock transcripts under STD and ECD throughout a circadian cycle as quantified by RNA-seqs. Data are presented as mean values ± SEM of 3 biological replicates. **f** Weight matrix of enriched cis-elements within 20 kb upstream of either STD or ECD rhythmic transcripts ranked by Normalized Enrichment Score (NES) and their associated transcription factors; Red star− rhythmic portion.

To seek clues for the regulator(s) of transcriptional re-writing, we compared the enrichment of cis-regulatory elements within 20Kb upstream of the transcription start sites in STD-rhythmic and ECD-rhythmic populations using iRegulon[29]. At the top of the list for the STD-rhythmic population (5502 transcripts) was BMAL1 (or ARNTL) and its binding site "ARNTL-DBD". This is a key transcription activator of the circadian feedback loop[30]. Such enrichment is therefore expected and a validation of the analysis. Transcription factors that bind to the top 5 STD-enriched cis-elements were ARNTL > ZFP143 > ZSCAN10 > CEBPG > YY2. In ECD-rhythmic population, transcription factors that bind to the highest enriched cis-elements were BHLHE40 > ETS1 > ZFP143 > HIF > ZBTB33 (Fig. 2f). BHLHE40, also known as DEC1, and its homolog BHLHE41 (or DEC2) are known circadian transcription factors that are also capable of evicting BMAL1 off its E-box binding sites, suppress BMAL1 transcription activation activity and response to environmental perturbation[31–34]. These proteins are thus promising candidates for being regulators of transcriptional re-writing. If DEC1/2 are indeed the transcription factors, it

would suggest that re-writing of circadian rhythmic transcripts after ECD involves removing of BMAL1 from its normal transcriptional activation targets concurrence with activating transcriptional activity of DEC1/2 on a different gene population. This would be a distinct mechanism from those underlaying circadian transcriptional re-pro-gramming/re-wiring associated with perturbations of metabolic process[35–37].

## Circadian whole-cell and nuclear proteomes are re-written by ECD

To see how ECD affects circadian gene expression at the translational and post-translational levels, we utilized the same strategy as for the transcriptional level, i.e., examining the circadian proteomes at whole-cell and nuclear compartment levels after ECD, then comparing them to the respective proteomes under STD. For after ECD, we quantified 5317 protein time series across both whole-cell and nuclear compartments. The fitness of these proteomes was validated by examining the enrichment of known nuclear proteins, the correlation between

compartmental proteomes and consistence of circadian patterns of WDR82 protein between mass-spec measurement and western blot detection across whole-cell and nuclear proteomes under both STD and ECD conditions (Supplementary Figs. 3b–d; Supplementary Data 3, Sheets 1,5).

At the whole-cell level, 909 (17%) series exhibited a circadian rhythmic pattern of abundance. This rhythmic population displayed a sharp bimodal distribution of peak abundance around 3 h before anticipated dark (CT9) or dawn (CT21). Their amplitude profile was not

significantly different with a $p$-value of 0.5002. Among 285 whole-cell STD rhythmic proteins, only 24 were also rhythmic after ECD (ROR$_{TE}$). 261 proteins, or 92% of STD rhythmic proteins, lose their rhythmicity (LOR$_{TE}$) while 885 proteins, which is about 3 times the number of STD rhythmic proteins, gained rhythmicity (GOR$_{TE}$) in response to ECD. (Fig. 3a, c, e; Supplementary Data 3, Sheets 1–4) Collectively, ECD re-writes the rhythmicity of 1146 proteins, constituting 98% of all rhythmic proteins under both conditions at the whole-cell protein level (Fig. 3f).

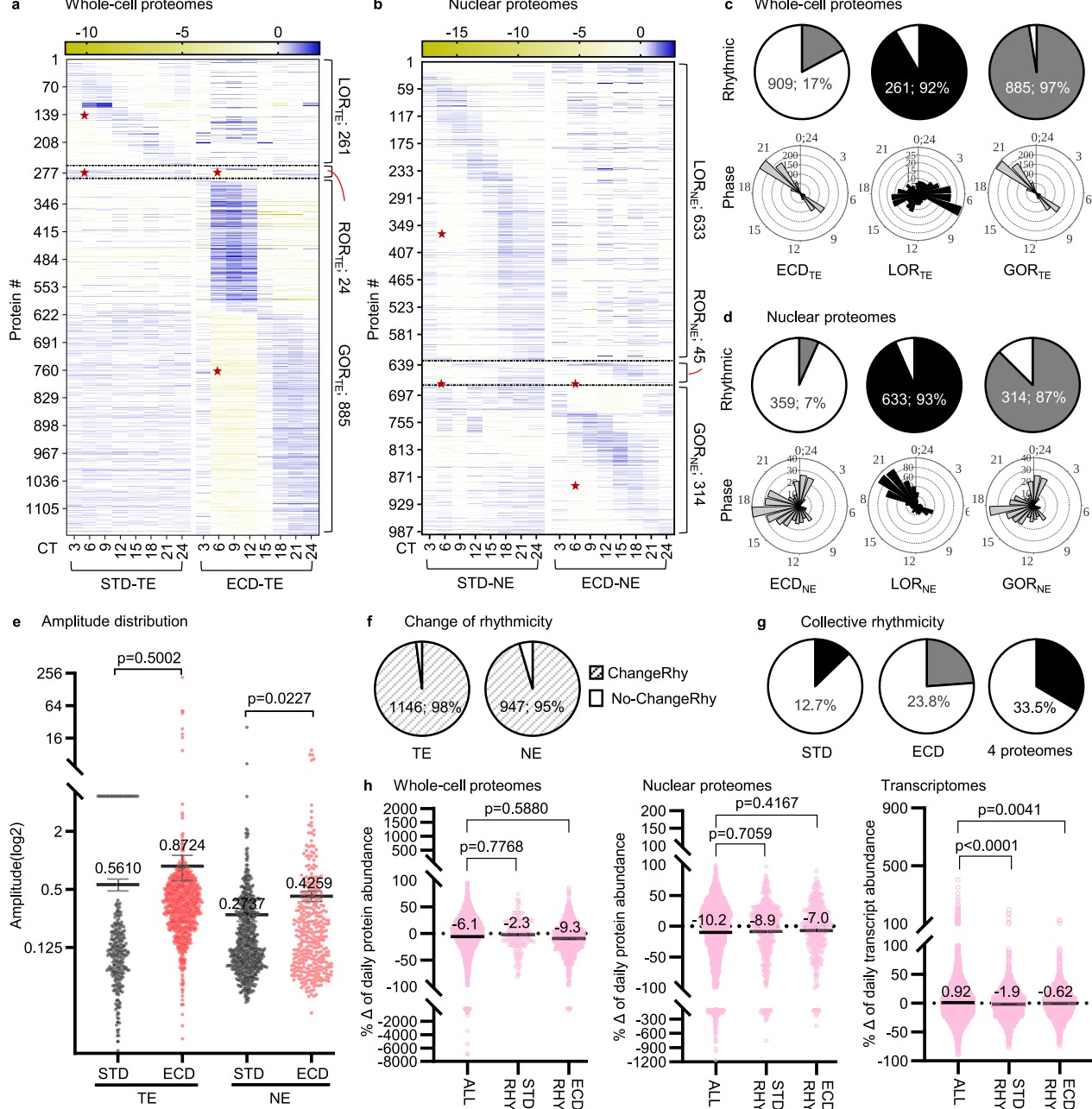

**Fig. 3 | ECD re-writes rhythmicity of circadian whole-cell (TE) and nuclear (NE) proteomes. a**, **b** Relative-abundance heatmap (normalized to series average; log$_2$ scale) of rhythmic proteins in whole-cell (**a**) or nuclear (**b**) proteomes under ECD in comparison with STD. **c**, **d** Percent rhythmic protein time series and the corresponding phase distribution of proteins that either loss rhythmicity (LOR), retain rhythmicity (ROR) or gain rhythmicity (GOR) in the whole-cell (TE) (**c**) or nucleus (NE) (**d**) circadian proteomes. **e** Comparative amplitude of all 4 circadian proteomes (log$_2$ scale). **f** Percent of all proteins that change their rhythmicity at the whole-cell or nuclear compartment levels. **g** Estimate percentages of rhythmic proteins, collectively. **h** Comparative distribution of change in cumulative daily abundance of whole-cell proteomes, nuclear proteomes or transcriptomes in response to ECD between all, STD rhythmic and ECD rhythmic populations. For (**e**, **h**), data are presented as population average ± SEM, with each dot being a member of the population. The $p$-values of (**e**, **h**) were derived from unpaired 2-tailed $t$ test. Red star−rhythmic portion. See also Supplemental Fig. 4.

In the nuclear compartment, 359 (7%) protein time series were rhythmic with their phases distributed throughout a large part of the cycle. The average amplitude of these proteins (0.4259) was significantly higher than their counter part under STD (0.2737). 633 of 678 STD rhythmic proteins (93.4%) lose their rhythmicity ($LOR_{NE}$) while 314 proteins acquired rhythmicity ($GOR_{NE}$) in response to ECD. There were 992 proteins, in total, that exhibit a rhythmic pattern of abundance in the nuclear compartment, of which 95% re-wrote their rhythmicity in response to ECD (Fig. 3b, 3d–f; Supplementary Data 3, Sheets 5–8). Collectively, 12.7%, 23.8%, and 33.5% proteins were rhythmic under STD, ECD, or both conditions, respectively (Fig. 3g).

To see if switching rhythmicity of the circadian proteomes associates with the level of protein expression, we examined the percent change of collective protein abundance over the entire circadian cycle between STD and ECD. We saw no significant difference in distribution profiles of the change percentages in the STD rhythmic population or the ECD rhythmic population in comparison to that of all proteins, at the whole-cell or nuclear compartment level. Similar analysis of the transcriptomes showed a significant difference between the STD rhythmic or ECD rhythmic transcript population with the all-transcript population (Fig. 3h). Thus, ECD re-writes the rhythmicity of proteins without changing the cumulative daily abundance, population wide. The difference in the association of switching rhythmicity with level of expression between circadian transcriptome and circadian proteomes might reflect the deviation in molecular underpinnings of their rhythmic accumulation: transcript abundance includes a large contribution from newly synthesized transcripts while protein abundance involves mostly processing, post-transcriptionally or post-translationally, after being synthesized.

Such high percentages of change in rhythmicity at the transcript and protein levels were beyond our anticipation. To see if loosening the stringency for rhythmic calling would have an impact on the change, we re-performed the analysis using one of the highest recall (RAIN)[38] or most popular (JTK-Cycle)[39] algorithm. Both showed insignificant or mild changes in the magnitude of re-writing of the proteomes or transcriptomes, respectively (Supplementary Fig. 4). These observations suggested that ECD re-wrote most of the protein rhythmicity in the whole-cell or nuclear compartment. The entire circadian gene expression process from transcription to post-translation was thus re-written by ECD.

The increase in protein rhythmicity at the whole-cell level, both quantity (from 3.9% to 17%) and synchrony (dispersed to discrete bimodal phase distribution), after ECD compared to STD is intriguing (Fig. 1b-TE vs Fig. 3c-$ECD_{TE}$). The circadian clock has evolved as a mechanism to help organisms synchronize their internal biological processes with anticipated changes of daily environmental cycles. One would expect a lessening, rather than increasing, of protein rhythmicity under environmental circadian disruption. However, biological systems are composed of interactive networks, which help their resilience to external perturbations by redistributing, in part, the impact. Under a circadian disrupted condition, these networks are likely compromised, making their nodes more vulnerable and thereby weakening their resilience. From this perspective, ECD could lead to more protein rhythmicity in a synchronous manner, which was what we observed. In agreement with this interpretation, the resilience of processes such as inflammation, glomerular and tubular injuries is compromised after ECD[6,40,41]. The gain of rhythmicity likely reflects not a strengthening of the system but a loss of resilience to perturbations.

## Post-transcription and post-translation are dominant contributors to circadian proteomes after ECD

Given that post-transcription and post-translation are the major contributors to protein rhythmicity under STD, we asked how ECD would affect those contributions. We examined overlaps between the circadian transcriptome, circadian whole-cell proteome, and circadian nuclear proteome after ECD as performed for STD. After ECD, we found 5028 transcripts of which whole-cell protein or nuclear protein were also quantified, constituting 24% quantified transcripts and 95% quantified proteins (Fig. 4a–c). Within the 5028 sub-population, 73% rhythmic whole-cell proteins did not have rhythmic transcripts (Fig. 4d–e; Supplementary Data 4). This percentage was substantially higher than the 52% under STD. In the nuclear protein rhythmic sub-population, 86% were not rhythmic at the whole-cell protein level (Fig. 4f, g). The rhythmicity of these proteins was acquired post-translationally ($GOR_{PTA\_ECD}$), similarly to the $GOR_{PTA\_STD}$ population. Thus, after ECD post-transcription was the major contributor to whole-cell protein rhythmicity, while post-translation processing was the dominant contributor to nuclear protein rhythmicity.

To seek further evidence for the contribution of post-transcription to circadian whole-cell proteomes, we examined two properties of rhythmicity, amplitude, and phase, in populations of genes in which both transcripts and whole-cell proteins were rhythmic under STD or ECD. In the gene expression process, transcripts, which have an average half-life of 4.8 min[42], are produced before proteins. Therefore, a rhythmic gene population with more contribution from transcription would have a higher ratio of genes with the phase of transcript leading the phase of protein to a population with the phase of transcript lagging the phase of protein. We found the ratio to be ~4:1 under STD but ~ 1:1 after ECD (Fig. 4h). Another prediction is that a rhythmic gene population with higher contribution from protein rhythmicity would have a higher ratio of protein's amplitude to transcript's amplitude. The amplitude ratio was ~1:4 under STD and increased to 9:1 after ECD (Fig. 4i). Thus, transcription contributes more to protein rhythmicity under STD than ECD while post-transcription contributes more to protein rhythmicity after ECD than STD. These results are consistent with our predictions and are additional pieces of evidence for a high contribution of post-transcription to protein rhythmicity after ECD.

## Nuclear protein import and export process associates with nuclear circadian proteomes

The nuclear protein import and export (PNIE) process is the major contributor to nuclear protein abundance. While much is known about the roles of this process in circadian rhythmic nuclear accumulation of a handful of core clock proteins[43–47], whether it plays a role in the accumulation of hundreds of other proteins in the nuclear circadian proteomes under STD or ECD remains unclear. As an initial step to answer this question, we test if there is an association between the nuclear circadian proteomes and the PNIE process. Nuclear localization and export sequences (NLS/NES) are key components of the nuclear import/export process[48] and are also intrinsic properties of cargo proteins. Nuclear proteins that also harbor NLS/NES sequences most likely utilize the PNIE process to transport into or from the nucleus. If there is an association between nuclear circadian proteomes and the PNIE process, the NLS/NES-containing nuclear rhythmic-only population should exhibit a stronger intra-functional network than the NLS/NES-containing whole-cell rhythmic-only population as the later population is less likely to utilize the PNIE process.

Using NLStramadus algorithm[49], we found 95 NLS or NES distributed among 46 proteins in the $GOR_{PTA\_STD}$ population, of which each member was rhythmic in the nucleus but not at the whole-cell level. The respective numbers were 34 NLS/NES and 23 proteins in the population of proteins that were rhythmic only at the whole-cell level (Supplementary Data 5, Sheets 1–2). Interestingly, STRING[50] analysis showed while the 46 NLS/NES-containing proteins in the $GOR_{PTA\_STD}$ population exhibited a high degree of functional interaction network with the protein-protein interaction (PPI) enrichment $p$-value of less than 1.0e−16, their counterparts in whole-cell rhythmic only population showed no enrichment of functional interaction network (PPI enrichment $p$-value = 1) (Fig. 5a, b). Similar results were also observed

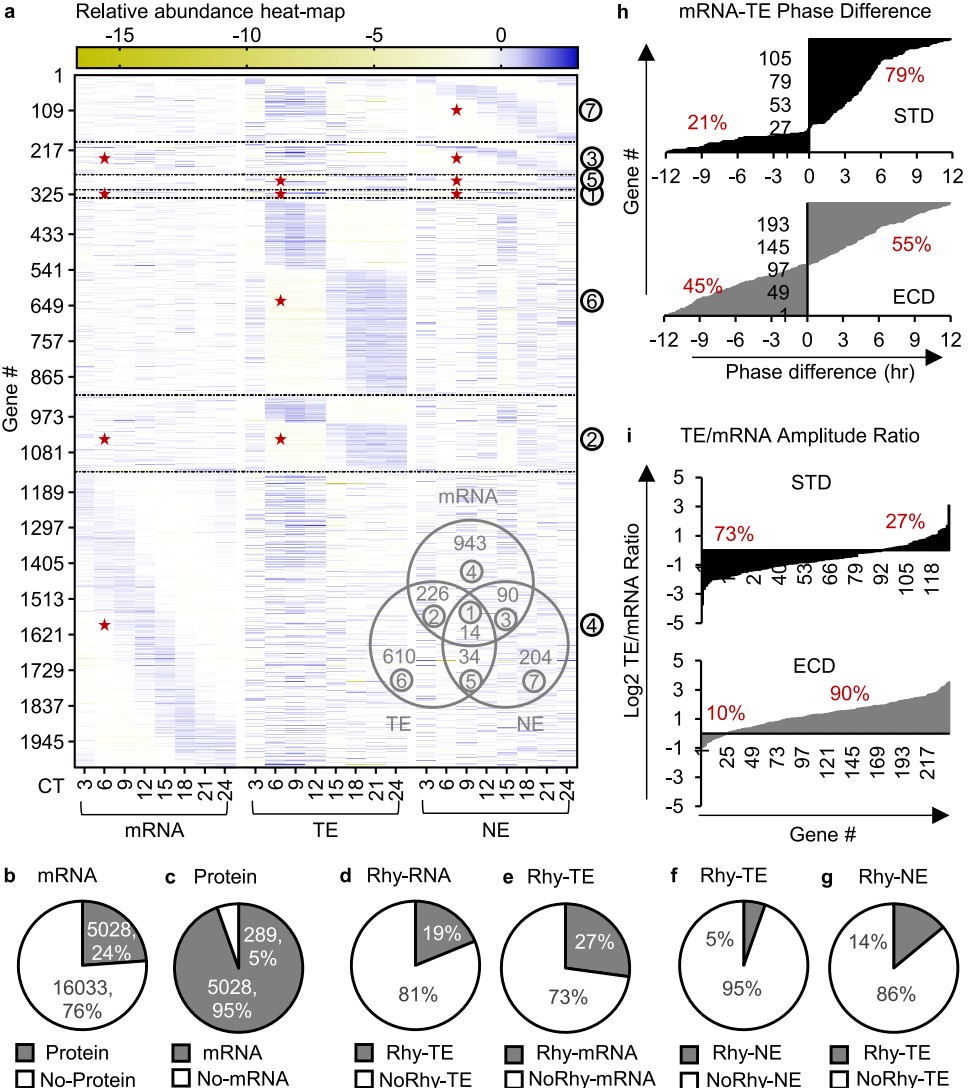

**Fig. 4 | Contributions of post-transcription and post-translation in circadian gene expression under ECD. a** Relative abundance heatmap (log₂) of rhythmic transcripts, rhythmic whole-cell protein and rhythmic nuclear proteins in 5028 genes of which both transcript and protein were quantified under ECD. Venn diagram shows overlaps between each population shown in (**a**). (**b, c**) Percentages of transcripts with or without proteins and vice versa. **d, e** Percentages of rhythmic transcripts with or without rhythmic whole-cell proteins and vice versa.

**f, g** Percentages of rhythmic whole-cell proteins with or without rhythmic nuclear proteins and vice versa. **h** Percentage of genes with the phase of transcript leads or lags the phase of whole-cell protein under STD or ECD. **i** Percentages of genes with the amplitude of whole-cell protein rhythms is lower or higher than that of the corresponding transcript rhythms under STD or ECD; Red star−rhythmic portion; Number in circle−corresponding groups between heatmap and Venn diagram in (**a**).

after ECD (Fig. 5c, d; Supplementary Data 5 Sheets 3–4). These results demonstrated a strong association between the nuclear protein import and export process with the nuclear circadian proteomes under both STD and ECD conditions. They do not, however, rule out the contribution of other processes such as protein degradation, which also partially happens in the nucleus[51].

Given that the association between nuclear circadian proteomes and the PNIE process was established, we then examined all proteins that were implicated in the process, looking for ones that are rhythmic in the nuclear compartment. Among 251 examined proteins, we found 5 (LMNA, PHB2, PIK3R1, PPP3CA, RAN) and 2 (CSE1L, SQSTM1) were rhythmic in the nuclear compartment under STD or ECD, respectively. Among these 7 proteins, RAN and CSE1L (also known as CAS) are of particular interest. While both are essential components of the PNIE process, RAN primarily regulates protein import, but CSE1L regulates protein export[48]. More importantly, in the nuclear compartment RAN was rhythmic under STD but lost its rhythmicity after ECD while the

reverse happened to CSE1L. Under STD, both were not rhythmic at the whole-cell level (Fig. 5f). These observations indicated that under normal conditions RAN gained rhythmicity upon nuclear entry, which was lost in response to ECD, while CSE1L only gained rhythmicity in the nuclear compartment in response to ECD. These proteins are thus strong candidates for being regulators of nuclear circadian proteomes under STD or after ECD, respectively (Fig. 5e-f).

## ECD re-writes circadian molecular functions

Since proteins are the functioning molecules of genes and there is a poor association between circadian rhythmicity of transcripts and proteins, we assessed the impact of ECD on circadian functions by performing analysis on the rhythmic whole-cell protein populations. Gene Ontology enrichment analysis of the ECD-rhythmic population alone showed that it is enriched with many known ECD-associated pathways, at direct term or child term, such as Interleukin-6 Production, C-type Lectin Receptor (Adaptive Immune System), KRAS

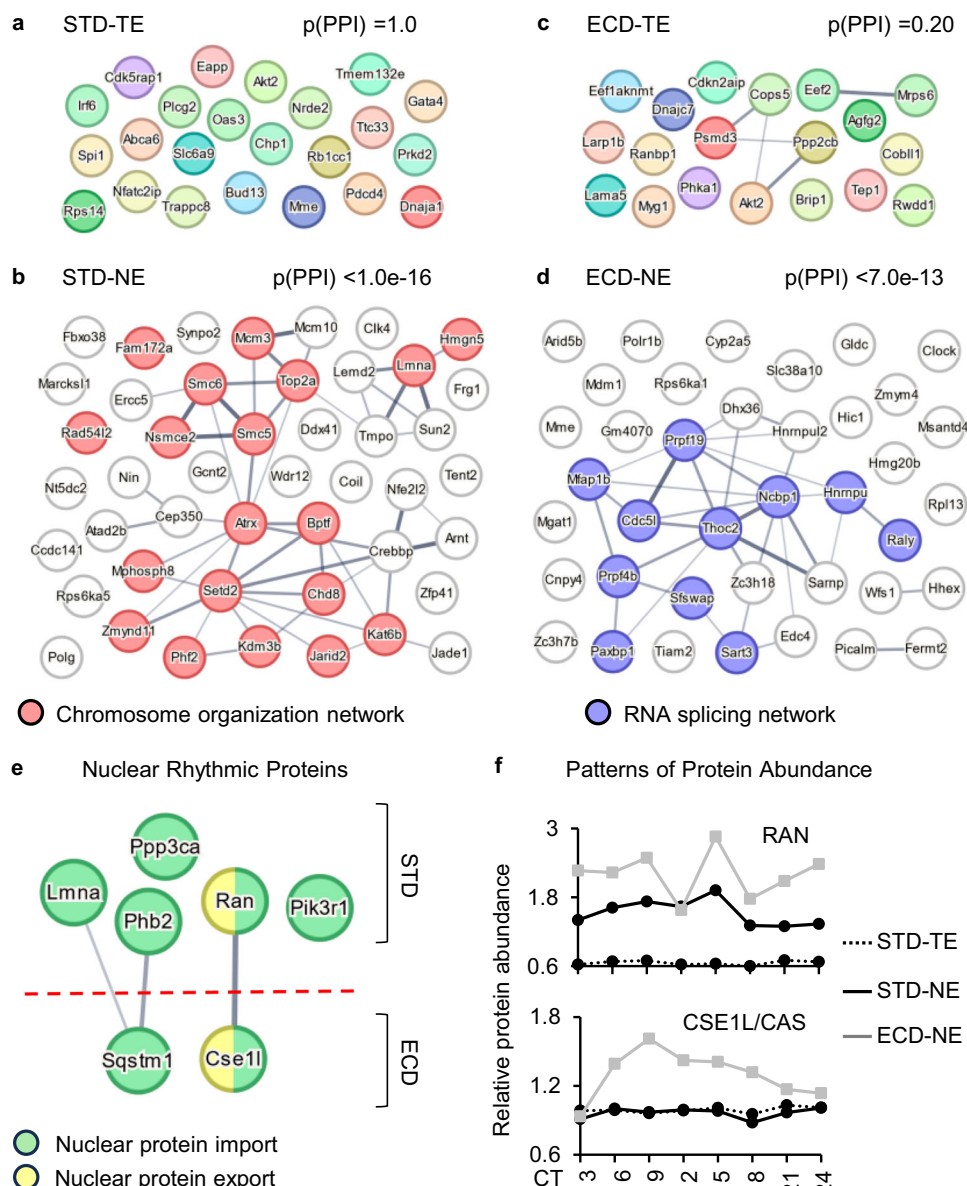

**Fig. 5 | Association of nuclear protein import/export with nuclear circadian proteomes. a–d** Functional protein interaction network analysis of NLS/NES-containing proteins that are rhythmic only at the whole-cell level (**a, c**) or in the nuclear compartment (**b, d**) under STD (**a, b**) or ECD (**c, d**) condition, using STRING algorithm. Line thickness is proportional to strength of evidence for the interaction. **e** PNIE-implicated proteins that are rhythmic in the nuclear compartment under STD or ECD. **f** Comparative protein abundance of RAN or CAS at the whole-cell and nuclear compartment level under STD and ECD throughout a circadian cycle. NLS/NES−nuclear localization/export sequence; PNIE−protein nuclear import/export process. PPI−protein-protein interaction network.

(Signaling by Receptor Tyrosine Kinase), TP53 (Signaling by FGFR) or GnRH (HIV 1 Infection)[41,52,53], as expected (Fig. 6a, italic). This population was also enriched with many processes, of which mRNA Splicing, Regulation of Mitochondrial Membrane Potential and Signaling by Receptor Tyrosine Kinases were on the top of the list. Intriguingly, there was an apparent day-night functional partitioning with unknown biological implications (Fig. 6a).

Comparative G.O. enrichment analysis between STD-rhythmic vs ECD-rhythmic whole-cell protein populations showed the enrichment of several STD-enriched processes that are important for circadian functions, such as Response to External and Biotic Stimuli, Digestion, Glucagon Signaling Pathway and Autophagy, was diminished in response to ECD. On the other hand, there were several other processes, including Adaptive Immune System, Peptide Metabolism, TP53 Regulated Transcription and various intra-cellular transportation,

gained in enrichment in response to the change in conditions (Fig. 6b). Most of these gain-in-enrichment processes were found in ECD enriched processes, consistent with a gain of rhythmicity as the major contributor to the circadian proteome.

Among proteins undergoing significant change in rhythmicity were regulators of these loss or gain of enrichment processes, including circadian-associated factors, such as PRMT5[54], SRSF6, XRCC5[55] (mRNA processing), RPLP0, EIF5, EIF4EBP2 (translation initiation), HRAS, AKT2[56] (Ras signaling), SEC61B, IPO9, EXOC6 (protein transport) (Fig. 6b, c). Furthermore, in our dataset we also found the expression of several genes that were implicated in circadian disruption exhibited a change in their pattern, at the level of whole-cell or nuclear protein, in response to ECD such as KRAS, HSF1, SREBF1, CTNNB1, NFIL3 and IRF8[7–9]. The change in circadian rhythmicity of these proteins not only affirms the goodness of our dataset but also

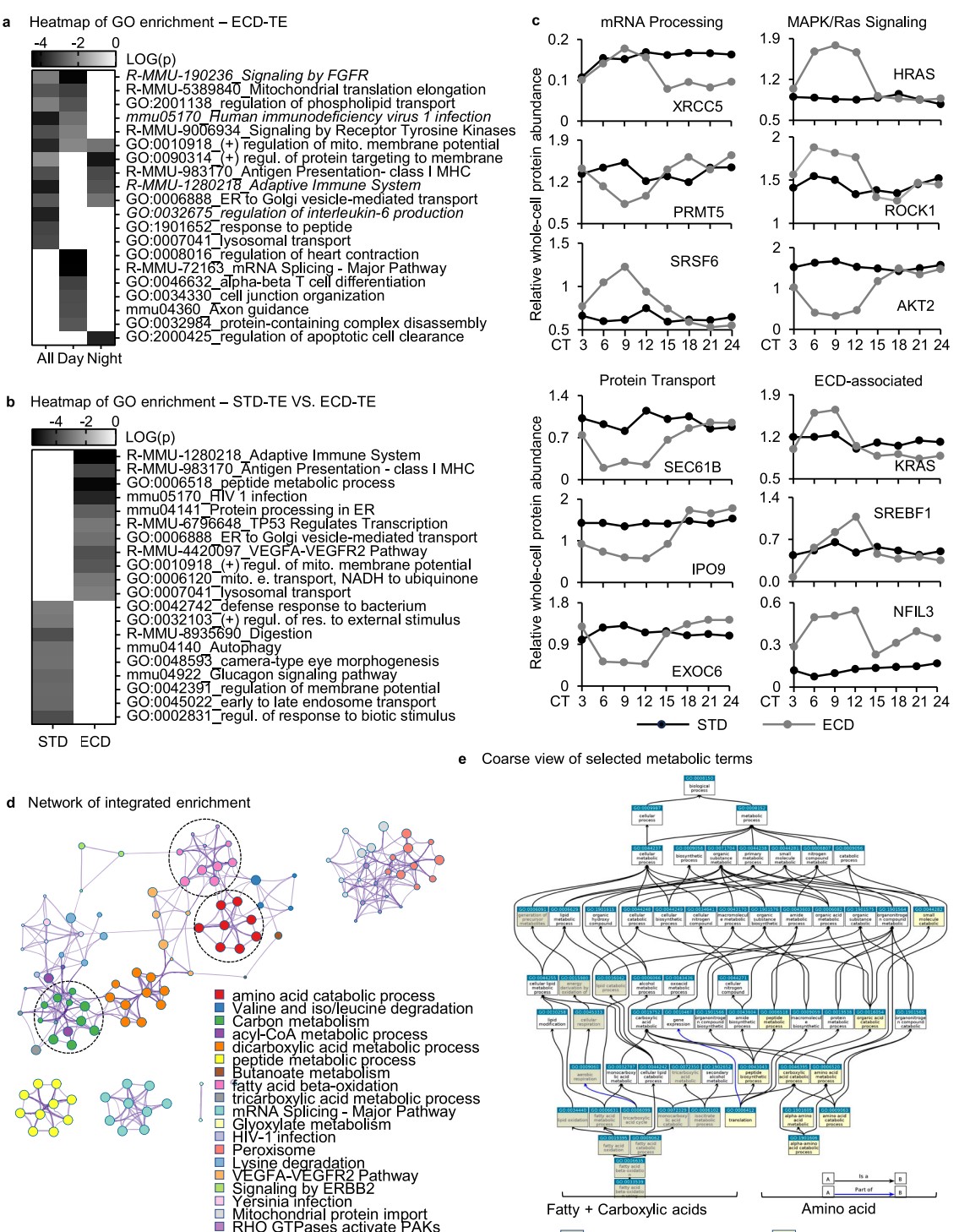

**Fig. 6 | Comparative functional G.O. enrichment analyses. a, b** Heatmap of G.O. enrichment analysis of (**a**) ECD whole-cell rhythmic proteins peaking during daytime, night-time or both, (**b**) whole-cell circadian proteome under ECD in comparison with STD. **c** Comparative whole-cell protein abundance throughout a circadian cycle of representative factors in mRNA processing, RTK/Ras signaling, protein transport and ECD-associated under STD versus ECD conditions. **d** Reactome enrichment analysis of (**b**). **e** G.O. Slim map of changes in enrichment of terms related to fatty acid, carboxylic acid and amino acid, as circled in (**d**), in response to ECD. A replicate of (**e**) with bigger font could be found in Supplemental Fig. S5. Italic−known ECD-associated pathways, direct term or child term.

offers avenues for deciphering the molecular underpinnings of circadian disruption. ECD thus re-wrote a wide range of circadian functions at the molecular level.

Given the central role of liver in metabolism, we took a closer look at the effect of ECD on metabolic functions. Analyzing the enrichment and network interaction of metabolic terms, we found that fatty acid and carboxylic acid metabolic processes lose most of their enrichment while amino acid metabolic processes, which were also enriched under STD, significantly increased in enrichment (Fig. 6d, e, Supplementary Fig. 5). These results suggest there is a change in prioritizing of circadian regulation from carbohydrate and fat to protein metabolism in response to ECD. The loss in timing of fat and carbohydrate metabolism might underlie, at least in part, ECD-associated metabolic disorders.

## Discussion

Although many studies were conducted on the effects of ECD, including Jetlag and shiftwork, and their molecular underpinnings, they mostly examined the association of ECD with some targeted biological dysfunctions. Here we took a comprehensive approach, interrogating the impact of ECD on the circadian gene expression process from transcriptional to post-translational levels as the point of entry to comprehensively study the biology of ECD. The approach proved its value as it yielded an unprecedented understanding of the impacts of ECD on circadian biology at the molecular level. It not only unveiled an astounding change in rhythmicity over nearly the entire circadian gene expression process with individual contribution of each stage, but also identified additional physiological processes that might be impacted by ECD. The closest observations that we could think of are re-programming and re-wiring of liver circadian transcriptome by high fat diet[37], caloric restriction[36] or lung adenocarcinoma[35]. In these previous cases, the change in re-programming and re-wiring was observed at the transcriptional level, but unknown at the protein levels, and was associated with perturbations of metabolic state of the cells, mediated by downstream clock-controlled processes. In this case, the re-writing was rooted in the perturbation of light information, impacted the whole gene expression process and likely acts on the core circadian clock feedback loop as well as post-translational processing in a distinct and complement mechanism to that of re-programing and re-wiring. Moreover, in addition to the re-writing of rhythmicity, we also found 95% of rhythmic transcripts and 94% rhythmic nuclear proteins do not have rhythmic whole-cell proteins. Taken together, these numbers indicated that while the change at transcriptional level might be the seeding event, it has little contribution to the associated change of circadian functional outputs in response to ECD.

It has generally been believed that it takes only a few days to recover from ECD. Such "understanding" is supported by results of behavioral studies in human and mouse. For instance, in human the recovery time for jetlag is about 1day/time zone[57,58]. A study of mouse model of ECD also showed daily temperature and running wheel rhythms were recovered after 1 week[41]. However, these studies were conducted under diurnal condition, in which the normal light-dark cycle was present. Light is known to produce a masking effect on animal behaviors and physiology. Thus, an apparently normal behavior under light-dark condition after few days of recovery might indicate an actual recovery or a passive reflection of light masking, at least in part. Here, under circadian (constant) condition we found that the entire circadian gene expression process is re-written even after 1 week of recovery, suggesting that ECD recovery did not happen after one week, at least at the molecular level. The discordance in recovery at molecular and behavioral levels might indicate a diversion or masking of behaviors by light. We favor the later scenario as these behaviorally "normal" mice are very susceptible to physiological challenges[40,41]. Further experiments are needed to test the validity of these two possibilities as well as to delineate the required time for ECD recovery at the molecular and behavioral levels. These pieces of information are crucial for proper interpretation of previous results and the design of future ECD studies.

In mammals, the circadian clock is established upon the transcription-translation feedback loop of less than a dozen core clock genes, which drives circadian functional outputs via rhythmic expression of hundreds to thousands of clock-controlled genes in each tissue. While contribution of post-translational controls such as protein phosphorylation, protein acetylation, protein nuclear localization, or protein turnover in the functional expression of core clock genes had been well-established[59,60], it is largely unknown for the clock-controlled genes. Our finding that 94% and 86% of nuclear rhythmic proteins are not rhythmic at the whole-cell level under STD or ECD conditions, respectively, is striking. Coupling these percentages with

48% under STD or 27% after ECD of rhythmic whole-cell proteins having rhythmic transcripts we could estimate that rhythmic transcripts contribute to 2.8% and 3.8% of rhythmic nuclear proteins under STD or ECD, respectively. The percentages suggested that post-translational processing is the dominant contributor to protein rhythmicity in cellular compartments, where most biological functions are performed. Thus, while the circadian clock is established on transcriptional regulation, its functional outputs are likely controlled at the post-translational level.

These results also beg the question of why cells generate thousands of rhythmic transcripts but use less than 5% for protein rhythms. This is puzzling. Gene expression is very sensitive to changes in extracellular conditions. While circadian-entrained environmental information (light, temperature) is constant under circadian condition, cells are still presented with numerous rhythmic extracellular signals generated within the organism via sleep, activities, endocrine signals, or cell-cell communication. As primary products of gene expression and with an average half-life of less than 5 minutes[42], thousands of transcripts, not surprisingly, were found accumulated in a circadian rhythmic manner. The large quantity might reflect, at least in part, the sensitivity of cells to multitude of extra-cellular inputs in combination with high dynamics due to low stability of transcripts. To efficiently extract useful information presented by such a large and diverted transcript population, cells would need a strategy to filter out transcriptional "noises" before passing them along for functioning. Post-transcriptional and post-translational processes could be "filters" that were evolved for this purpose. In this view, transcript rhythms represent the collective sensitivity of cells to changes in extra-cellular inputs while protein rhythms represent the filtered functional responses. Additionally, proteins are very stable, with an average half-life of ~3.6 days in mouse liver[61]. This is about 1000x more stable than transcripts and significantly longer than a circadian cycle, resulting in a damping of protein rhythms. These characteristics provide a possible explanation, at least in part, for the difference in the numbers of rhythmic transcripts versus rhythmic proteins.

In summary, our study provided a comprehensive understanding of the circadian gene expression process across transcriptional, translational and post-translational levels under normal and ECD conditions. It showed that environmental circadian disruptions re-writes circadian gene expression system and molecular circadian functions, which are not recovered, at least at the molecular level, after one week of recovery. The study also unveiled a crucial role of post-translational processing in circadian gene expression and function and paved multiple roads for deciphering the molecular underpinnings of ECD.

## Methods

### Tissue collection

All animal experiments were approved by the Institutional Animal Care and Use Committee (IACUC) at the Morehouse School of Medicine. Throughout the experiment, mice were fed ad lib with normal chow (PicoLab, CAT: 3005659-220) and housed in standard mouse cage with a maximum of 4 mice/cage. Cages were changed every other week. 48 wildtype (C57BL6/J) male mice at around 12-week old were purchased from Jackson Laboratory. They were acclimated to our facility environment before randomly divided into 16 groups of 3. Eight group were subjected to standard light-dark cycle (STD; ON at 6AM and OFF at 6PM) while the remaining groups were subjected to Environmental-Circadian Disruption light-dark regiment[13] (ECD; 4 cycles of 6-hr phase-advance light shift). After the last phase shift, ECD animals recovered under STD condition for 1 week. All animals were then subjected to constant darkness (DD). Starting from the 2nd day in DD, all 3 animals from a group were euthanized followed by decapitation, in accordance with our approved IACUC protocol, at the desired time point. Tissues were collected individually. A small piece of each tissue was preserved

in RNA later solution (Invitrogen) and the rest was flashed frozen in liquid N$_2$ for downstream RNA or protein analysis, respectively.

### RNA-seq

RNA-seq analysis was performed by the Emory Genomic core for each of the 48 tissue samples using PolyA+ (mRNA) at 100 million reads (50 M paired ends). Alignment to mm39 mouse genome was performed using STAR v2.5.2 with 76.2-88.1% uniquely aligned reads. After filtering out time series with an abundance of less than 50 transcripts or detection in less than 32 of 48 samples, we successfully quantified 21166 transcript time series across both STD and ECD conditions.

### Mass spectrometry

Mass spectrometry analysis was performed by the Thermo Fisher Scientific Center for Multiplex Proteomics at Harvard Medical School on an Orbitrap Eclipse mass spectrometer. Whole-cell and nuclear extracts from each of the 48 livers were prepared independently as previously described[62,63]. The triplicates were pooled, in equal mass, according to group and compartment. Approximately 50 µg of each pooled sample was then subjected to 18-plexes, with 2 spike samples in each 18-plex, for parallel analysis. The 18-plexes were normalized using the average of the 2 spike samples. At 1% False Discovery Rate (FDR) of peptide spectral match and 5% FDR of protein identification, the analysis identified 8,294 proteins in extracts under ECA condition and 6,261 proteins in extracts after ECD condition. Of these identified proteins 5257 to 7314 protein time series were quantified for each of the 4 proteomes and were used for further analysis.

### Circadian rhythmic detection

To detect circadian rhythmicity, we used the common denominator of two algorithms: RAIN and BIO_CYCLE. RAIN is a non-parametric algorithm that detects both symmetric and asymmetric waveforms with high reproducibility and recall while BIO_CYCLE is a parametric deep learning algorithm that could detect various waveforms with low rate of false positive and high precision[38,64,65]. With this strategy, we hope to combine the strength of the algorithms to achieve high recall with low false positive in rhythmic calling. An abundance time series was scored significantly rhythmic only if its q(RAIN) and q(BIO_CYCLE) are both less than 0.05 for transcript or p(RAIN) and p(BIO_CYCLE) are both less than 0.05 for protein time series. JTK-CYCLE was also used as a more stringent algorithm[39].

### Gene ontology and cis-element enrichment analysis

G.O. analysis was performed using Metascape[66] with the background is all proteins that were quantified in this study.

### Cis-element enrichment analysis

This was performed using iRegulon algorithm[29] version 1.3 with the following parameters: 20 kb upstream of the transcription start site; 10 K (9713 PWMs) motif collection; 0.03 ROC threshold for area under the curve; 0.001 FDR.

### Nuclear import/export sequences and protein–protein interaction network

NLS/NES sequences and their harbored proteins in each rhythmic population were identified using the NLStradamus algorithms[49]. NLS/NES-containing proteins were then subjected to Protein-protein interaction network analysis using String[50].

### Statistical analyses and graphs

Student's *t* tests, Pearson correlations and heatmaps were calculated using Graphpad PRISM 10. All other statistically analyses including those for circadian rhythmic detection, G.O. enrichment, cis-element enrichment, NLS/NES enrichment and Protein-Protein interaction network were performed using embedded calculation in each respective algorithm as shown above.

### Antibodies and RT-qPCR ProbesAntibodies

α-BMAL1 was a gift from Charles J. Weitz (Harvard Medical School); α-CLOCK (Abcam; ab3517); α-PER2 (Millipore; AB2202); α-NR1D1 (Abclonal; Catalog # A20452); α-SAP155 (Bethyl Laboratories, Inc; Catalog # A00-996A)

Probes:

| # | Name | Sequence |
|---|------|----------|
| 1 | mARNTL-qPCR-F | CAGAAGCAAACTACAAGCCAACA |
| 2 | mARNTL-qPCR-R | GGTCACATCCTACGACAAACA |
| 3 | mCLOCK-qPCR2-F | CCTTCAGCAGTCAGTCCATAAA |
| 4 | mCLOCK-qPCR2-R | CATGCCTTGTGGAATTGGTAAAT |
| 5 | mPER1-qPCR1-F | CCTGGAGGAATTGGAGCATATC |
| 6 | mPER1-qPCR1-R | CCTGCCTGCTCCGAAATATAG |
| 7 | mPER2-qPCR-F | CAACAACCCACACACCAAAC |
| 8 | mPER2-qPCR-R | CTCGATCAGATCCTGAGGTAGA |
| 9 | mPER3-qPCR-F | CACTCCAGGATGTGTGTTTCT |
| 10 | mPER3-qPCR-R | GATCTTCTGGGTGCAAGTATGT |
| 11 | mCRY1-qPCR-F | CTCAGTCCTTATCTCCGCTTTG |
| 12 | mCRY1-qPCR-R | CCACAGGAGTTGCCCATAAA |
| 13 | mCRY2-qPCR-F | GAGAACCATGACGACACCTATG |
| 14 | mCRY2-qPCR-R | AGCTTCTGTCTCTCCTCCTT |
| 15 | mCSNK1D-qPCR-F | TACTTCAACCTGGGCTCTCT |
| 16 | mCSNK1D-qPCR-R | CAATGGGAGTGGACATCTTCTT |
| 17 | mDbp-qPCR-F | CTGAGGAACAGAAGGATGAGAAG |
| 18 | mDbp-qPCR-R | TGGTTCTCCTTGAGTCTTCTTG |
| 19 | mLrwd1-qPCR-F | TGAAGAAGCTGAGGGAACTTG |
| 20 | mLrwd1-qPCR-R | TGTTGGTACAGCGAAGGATG |
| 21 | mCyc1-qPCR-F | ATTGCGAGAAGGCCTCTATTT |
| 22 | mCyc1-qPCR-R | TGCCATCATCATACTCCAAGAC |

### Reporting summary

Further information on research design is available in the Nature Portfolio Reporting Summary linked to this article.

## Data availability

All data supporting the findings of this study are available within the paper and its Supplementary Information. Original data is available in NCBI via accession number PRJNA1107923 and PRIDE via accession number PXD052085. Source data are provided with this paper.

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

## Acknowledgements
The authors thank Dr. Peter R. MacLeish and Dr. Carl H. Johnson for their valuable comments and suggestions in the preparation of the manuscript. We also thank many colleagues who gave valuable comments and suggestions during our presentation of the study at the GRC-Chronobiology 2023. We would also like to thank the following funding agencies: The study was supported by NIGMS grants 1 R16 GM146703-01 (H.D.), SC1 GM135112-01A1 (K.B.), H.D. is supported by NIGMS grant 1R16GM146703-01 and NEI grant R01EY026291, K.B. is supported by NIGMS grant SC1 GM135112-01A1, G.T. is supported by NEI grants R01EY026291 and R21EY031821, J.P.D. is supported by NIGMS grant GM127044, A.J.D. is supported by NIGMS grant R35GM136661, C.E. is supported by NIGMS grant GM127260, MSM Core facilities is supported by NIH grant 2U54MD007602, and Emory Integrated Genomics Core Facility is supported by grant RRID:SCR_023529.

## Author contributions
H.D. incepted the project and analyzed the data. H.D. designed the experiments with input from G.T., A.J.D., J.P.D., and M.P. H.D. performed the experiments with help from K.B. and C.E. H.D. wrote the manuscript with input from all authors.

## Competing interests
The authors declare no competing interests.
