## [Peer Review File · Nature Communications]

Environmental circadian disruption re-writes liver circadian proteomesREVIEWER COMMENTS

Reviewer #1 (Remarks to the Author):

This is a nice integration of the three data sets. Although transcriptome/proteome divergence has been noted before as has the greater oscillation of PTMs vs proteins in MS datasets, a fully integrated analysis including temporal perturbation is new. The most striking aspect of the data is that oscillating transcripts are much more common than oscillation of the proteome and indeed that those proteins that do oscillate derive from transcripts that don't.

Given what we know about sexual dimorphism in the response of the transcriptome, proteome and metabolic function to circadian disruption have the authors any information in this regard?

Besides comparing the whole cell and nuclear transcriptome have they looked at any specific PTMs?

Finally alterations in function are inferred indirectly by GO analysis but did they look at any actual functions, perhaps reflective of these pathways?

Reviewer #2 (Remarks to the Author):

Circadian clocks control large amounts of molecular and cellular events. Extensive amounts of work have been done to understand what genes are under the circadian control mostly under classic circadian conditions. How environmental alterations affect circadian gene expression remains poorly understood. To address this question, the authors in this study utilized environmental conditions to phase-shift the mammalian clock and then measure mRNA and protein abundance from livers collected from treated mice. It is interesting that upon the treatment, a considerable number of arrhythmic genes become rhythmically expressed while many regular circadian genes lost their rhythmicity. The abundance of many nuclear proteins becomes rhythmic, though their mRNA abundance is not, which suggests post-translational mechanisms control this process. Functional analyses reveal these clock-controlled genes and proteins participate in various cellular processes. The

manuscript is well written, and the evidence provided is solid. The findings are valuable complements to the existing big data of circadian transcriptome and proteome that were acquired under free-run conditions.

Comments

For rhythmic genes and proteins only seen by ECD, which ones were reported by prior papers; which ones are new? These should be compared and noted.

Fig. 1B why is the phase of most rhythmic proteins in the nucleus under ECD ~CT21?

Fig. 2 The condition used to disrupt the endogenous clock is "4 cycles of 6-hr phase-advance light shift". Correspondingly, circadian rhythmic genes are supposed to be advanced by 6hrs. However, it is surprising that many rhythmic genes lost their rhythmicity including transcripts, total protein, and nuclear protein while others gain it. How is this achieved? The transcription of core clock genes looks similar in STD and ECD samples (Fig. 2E), but 3118 genes lost their rhythmicity (Fig.2A).

Fig.3 For the rhythmic proteins only under ECD, were they reported previously? In 3D and 3E, "ECD" should be "ROR"?

Line 290 "Protein synthesis and turnover happen in the cytoplasm." Protein degradation does occur in the nucleus (Gardner et al., 2005 Cell). Is expression of components of the protein degradation machinery rhythmic in the nucleus under ECD?

Fig.4H, why do ~20% genes become phase-delayed under the phase advance condition?

Fig. 5, is there any functional test for the importance of the identified proteins involved in protein nuclear traffic? Is there a phase correlation between circadian expression peaks of the nuclear transporters and nuclear importing peaks of the ECD-induced rhythmic proteins?

Fig. 6E, words in boxes are too small to read.

For RNA-seq, were reads from splicing isoforms counted? For mass spec, were posttranslational modified peptides counted? The raw RNA-seq and mass spec data should be deposited in public databases.

REVIEWER COMMENTS

Due to formatting requirements of Nature Communication we made several formatting changes such as converting all secondary subheadings into primary subheadings, converting capital letters on figure panels to lower case, adding "Here we showed"... to the manuscript. However, we DID NOT highlight these changes in the manuscript as they don't impact the scientific content but might become a distraction.

Otherwise, all changes are highlighted in red color.

Reviewer #1 (Remarks to the Author):

This is a nice integration of the three data sets. Although transcriptome/proteome divergence has been noted before as has the greater oscillation of PTMs vs proteins in MS datasets, a fully integrated analysis including temporal perturbation is new. The most striking aspect of the data is that oscillating transcripts are much more common than oscillation of the proteome and indeed that those proteins that do oscillate derive from transcripts that don't.

1.1. Given what we know about sexual dimorphism in the response of the transcriptome, proteome and metabolic function to circadian disruption have the authors any information in this regard?

This is a great question, and we are also very interested in it. Recent studies showed a difference in response to circadian misalignment between males and females (Anderson et al. *Science Translational Medicine* 2023; Pariollaud et al *Science Advances* 2022; Chun et al. *Science Advances* 2022 - Ref # 9-11 in the text). Anderson et al showed profound changes in daily rhythms at the molecular (transcripts, proteins, gut microbiome) and behavioral (running wheel activity) levels in males but much less in females in response to shiftwork. It would be interesting to see if this is also the case in our study model and what the molecular mechanism of this resilient is. Such information will not only expand our understanding of sexual dimorphism, but also might open a door for potential therapeutic avenues for ECD. We are hoping for the means to conduct this study in the near future.

1.2. Besides comparing the whole cell and nuclear transcriptome have they looked at any specific PTMs?

This is another great question, going right along with what we planned for the very next stage of our study.

Given the known crucial roles of PMTs such as phosphorylation, Arginine methylation, acetylation... in the regulation of nuclear protein import/export process, and our finding potential roles of RAN and CAS (CSE1L) in nuclear circadian proteomes in this manuscript, changes in circadian PMTs might contribute to the formation and re-writing of nuclear

circadian proteomes in response to ECD. We thus planned to couple the analysis of PMTs with potential roles of RAN and CAS as a project in the very next stage of our study.

1.3. Finally, alterations in function are inferred indirectly by GO analysis but did they look at any actual functions, perhaps reflective of these pathways?

We haven't investigated actual functional alterations ourselves. However, few of our GO-implicated functional alterations, as well as some of their regulators, were observed in previous study. For example, pathways such as C-type lectin receptor, Toll-like receptor, GnRH signaling, KRAS signaling, which were perturbed in our G.O. enrichment analysis at direct term or child term, were previously associated with ECD perturbations. We revised Figure 6 as well as the accompanying text under "ECD re-writes circadian molecular functions" on pages 6-7 of the main text.

Reviewer #2 (Remarks to the Author):

Circadian clocks control large amounts of molecular and cellular events. Extensive amounts of work have been done to understand what genes are under the circadian control mostly under classic circadian conditions. How environmental alterations affect circadian gene expression remains poorly understood. To address this question, the authors in this study utilized environmental conditions to phase-shift the mammalian clock and then measure mRNA and protein abundance from livers collected from treated mice. It is interesting that upon the treatment, a considerable number of arrhythmic genes become rhythmically expressed while many regular circadian genes lost their rhythmicity. The abundance of many nuclear proteins becomes rhythmic, though their mRNA abundance is not, which suggests post-translational mechanisms control this process. Functional analyses reveal these clock-controlled genes and proteins participate in various cellular processes. The manuscript is well written, and the evidence provided is solid. The findings are valuable complements to the existing big data of circadian transcriptome and proteome that were acquired under free-run conditions.

Comments

2.1. For rhythmic genes and proteins only seen by ECD, which ones were reported by prior papers; which ones are new? These should be compared and noted.

There has been a very limited number of reports on analysis of "circadian" transcriptome or proteome under circadian disrupted condition. Among three transcriptomes and one whole-cell proteome of mouse liver that we are aware of, two were performed at either one circadian time point (Inokawa et al, *Sci Rep* 2020) or 3 Zeitgeber time points (Kettner et al *Cancer Cell* 2016). For the remaining transcriptome as well as the whole-cell proteome, the animals were subjected to a drastically different circadian disrupted protocol than ours and the tissues were collected under Zeitgeber time (Anderson et al *Sci Transl Med* 2023). These limitations made us hesitate to compare our findings with the findings in these studies, at the -omics level. We thus focused on the comparison at the level of individual genes. We

revised Figure 6 as well as the accompanying text under “ECD re-writes circadian molecular functions” on pages 6-7 of the main text to reflect these changes.

#2.2. Fig. 1B why is the phase of most rhythmic proteins in the nucleus under ECD ~CT21? We assume that there is a typo, STD instead of ECD, in this question.

Highly synchronous unimodal phase distribution is an intriguing feature of the STD nuclear circadian proteome that we also noticed but haven't had any evidence for how it comes about or what its functional implications are. Such a phase distribution was also observed in the circadian proteome of another cellular compartment, namely the mitochondria, but at a different circadian time, CT 4 (Neufeld-Cohen et al 2016, PNAS). As discussed in # 2.7, there is an association between rhythmic nuclear abundance of RAN and CAS with phase distribution of STD rhythmic nuclear proteome or ECD rhythmic nuclear proteome, respectively. Hopefully our investigation of RAN and CAS in the next stage of our study will shed some light into this question.

#2. 3. Fig. 2 The condition used to disrupt the endogenous clock is “4 cycles of 6-hr phase-advance light shift”. Correspondingly, circadian rhythmic genes are supposed to be advanced by 6hrs. However, it is surprising that many rhythmic genes lost their rhythmicity including transcripts, total protein, and nuclear protein while others gain it. How is this achieved? The transcription of core clock genes looks similar in STD and ECD samples (Fig. 2E), but 3118 genes lost their rhythmicity (Fig.2A).

This is another great question, zeroing on molecular underpinnings of re-writing from transcription to post-translation.

For the rhythmic transcripts, analyzing the upstream regulatory region we found that the STD rhythmic transcript population is enriched with binding site for BMAL1, as expected, while the ECD rhythmic transcript population is enriched with binding site for DEC1. Given that DEC1 is a known circadian transcription factor and is able to repress the transcriptional activation activity of BMAL1 (Figure 2 f, page 4, paragraph 1 of the main text), the switch in activity of those transcription factors in response to ECD could underlay the circadian transcriptome re-writing. We plan to investigate this hypothesis as a part in the next stage of our study.

For the rhythmic nuclear proteins, we found association of nuclear protein import/export process and changes in nuclear rhythmicity of RAN and CAS, two key components of the process, with re-writing of nuclear circadian proteome. RAN, which is a key regulator of nuclear protein import, is rhythmic under STD and loses rhythmicity under ECD while CAS, which is a key regulator of nuclear protein export, is not rhythmic under STD and gains rhythmicity under ECD. The changes in nuclear rhythmicity of these proteins might contribute to re-writing of the nuclear circadian proteomes. As part of the next stage in our study, we will investigate this hypothesis.

For the rhythmic whole-cell proteins, we found NO association of change in their rhythmicity with that of transcript's rhythmicity. This result points to mRNA processing, protein synthesis or protein turn-over as potential contributor(s) for re-writing of whole-cell protein rhythmicity. Consistence with this interpretation, our G.O. enrichment analysis of circadian whole-cell proteome showed mRNA processing and protein synthesis are among the highest enriched terms. We have yet singled out candidate(s) for further investigation.

#2.4. Fig.3 For the rhythmic proteins only under ECD, were they reported previously? In 3D and 3E, "ECD" should be "ROR"?

The populations presented in figures 3d and 3e are ECD, not ROR. We utilized these figures as the opportunity to present the distribution patterns of phase and amplitude of the proteomes rather than only ROR population. Besides, there are very small numbers of STD rhythmic proteins retaining their rhythmicity under ECD, 24 and 45 in whole-cell or nuclear proteome, respectively.

Some of the rhythmic proteins only under ECD were previously reported. Please see #2.1, #1.3 and section "ECD re-writes circadian molecular functions" on pages 6-7 of the main text for more details.

#2.5. Line 290 "Protein synthesis and turnover happen in the cytoplasm." Protein degradation does occur in the nucleus (Gardner et al., 2005 Cell). Is expression of components of the protein degradation machinery rhythmic in the nucleus under ECD?

Thanks for the reference and the question. We did not find protein degradation in the top enriched G.O. terms. Although we found 9 members of the Regulation of Protein Stability gene ontology (GO:0031647), including NCLN, WFS1, HSPD1, TARDBP, NR1D1, SREBF1, RNF149, ASGR2 and PDCD10, statistically exhibited a rhythmic pattern of nuclear abundance, only RNF149 ($p = 0.018$ and Amplitude = 0.18) was directly implicated in protein degradation. We revised the text to reflect this potential contribution of protein degradation (page #6 paragraphs #1&2).

#2.6. Fig.4H, why do ~20% genes become phase-delayed under the phase advance condition?

Thanks for the question. It made us realize that we did a poor job at titling this figure. This is the phase difference between transcript and whole-cell protein of the same gene under the same condition, not the phase difference due to ECD. Thus, ~20% is the percent of genes which have the phase of their rhythmic whole-cell protein leading the phase of their rhythmic transcript. We changed the title of the figure to "mRNA-TE Phase Difference" as an effort to mitigate this issue.

#2.7. Fig. 5, is there any functional test for the importance of the identified proteins involved in protein nuclear traffic? Is there a phase correlation between circadian expression peaks of

the nuclear transporters and nuclear importing peaks of the ECD-induced rhythmic proteins?

As stated in question #2.3, we are designing experiments to test whether RAN and CAS contribute to the formation as well as re-writing of the nuclear circadian proteomes in the next stage of our study.

We hypothesize that RAN contributes to importing of rhythmic nuclear protein abundance under normal (STD) condition while CAS contributes to exporting of rhythmic nuclear protein abundance under ECD condition. If this hypothesis is correct, we would expect the peak of RAN is about or little earlier than the peaks of rhythmic nuclear proteins that it imports. Our data showed that nuclear RAN peaks around CT15 while most rhythmic nuclear proteins peak between CT19 to CT22 under STD. Another prediction is that we would expect the pattern of nuclear CAS to be in antiphase with the patterns of rhythmic nuclear proteins that it exports. Under ECD, nuclear CAS peaks around CT9 when most rhythmic nuclear proteins are lowest (Figures 1d, 3d, 5f). Our observations are thus consistent with the predictions. At face-value, there seems to be a correlation between the peaks of nuclear transporters and rhythmic nuclear proteins, but this needs to be tested further.

#2.9. Fig. 6E, words in boxes are too small to read.

Due to space limitation, we were not able to present this figure in a readable font. We thus used this panel to provide a global view. An identical figure with a bigger font is presented as Figure S5. Sorry that we did not make this clear in the legend of this figure. We revised the figure legend, which include: "... A replicate of (e) with bigger font could be found in supplemental figure S5".

#2.10. For RNA-seq, were reads from splicing isoforms counted?

Yes, the presented RNA abundance includes all isoforms.

#2.11. For mass spec, were posttranslational modified peptides counted?

Yes, the presented protein abundance was cumulative.

#2.12. The raw RNA-seq and mass spec data should be deposited in public databases.

Yes, all raw data files were deposited in a public database and will be available when the paper is published. Accession numbers were included in the manuscript under Data availability.

REVIEWERS' COMMENTS

Reviewer #1 (Remarks to the Author):

My concerns are addressed albeit by indicating that they will be issues addressed in the next effort.

Reviewer #2 (Remarks to the Author):

The revised manuscript has addressed my questions raised for the earlier version; this reviewer does not have further questions.